# A Systematic Analysis of 3D Deformation of Aging Breasts Based on Artificial Neural Networks

**DOI:** 10.3390/ijerph20010468

**Published:** 2022-12-27

**Authors:** Jun Zhang, Ruixin Liang, Newman Lau, Qiwen Lei, Joanne Yip

**Affiliations:** 1School of Fashion and Textiles, The Hong Kong Polytechnic University, Hung Hom, Hong Kong, China; 2Laboratory for Artificial Intelligence in Design, Hong Kong Science Park, New Territories, Hong Kong, China; 3School of Design, The Hong Kong Polytechnic University, Hung Hom, Hong Kong, China

**Keywords:** breast skin deformation, backpropagation artificial neural network, gray relational analysis, computer-aided system

## Abstract

The measurement and prediction of breast skin deformation are key research directions in health-related research areas, such as cosmetic and reconstructive surgery and sports biomechanics. However, few studies have provided a systematic analysis on the deformations of aging breasts. Thus, this study has developed a model order reduction approach to predict the real-time strain of the breast skin of seniors during movement. Twenty-two women who are on average 62 years old participated in motion capture experiments, in which eight body variables were first extracted by using the gray relational method. Then, backpropagation artificial neural networks were built to predict the strain of the breast skin. After optimization, the R-value for the neural network model reached 0.99, which is within acceptable accuracy. The computer-aided system of this study is validated as a robust simulation approach for conducting biomechanical analyses and predicting breast deformation.

## 1. Introduction

According to World Population Prospects 2022 [1], 10% of the global population is 65 years old or over, among whom 55.7% are women. Because of the female advantage in life expectancy, women outnumber men at older ages in almost all populations. With this demographic pressure [2], the health issues of senior women have been a topic of great concern. Therefore, they are increasingly encouraged to partake in sporting activities to improve both their physical and mental health [3,4]. However, physical activities may cause discomfort and pain for women due to excessive breast movement [5]. Anatomically, the breasts are internally supported by only Cooper’s ligaments and breast skin [6], which easily leads to large movements of the breasts. Previous studies claim that large vertical deformations (4.2–9.9 cm) of the breasts are produced during running on a level treadmill in a constrained laboratory environment [7,8,9,10,11,12]. In fact, breast deformation may vary significantly with different body movements. For senior women, ageing reduces the elasticity of the skin and fibers within the breasts [13]. Compounded with breast feeding and menopause, aging breasts suffer from increased breast clinical manifestations and motion-related mastalgia [14]. Additionally, body flexibility, walking speed, and balance ability are reduced with age [15,16]. Remedies might include mild exercises, such as yoga and tai chi, which are also more suitable for the elderly [17].

Another limitation of current research on breast biomechanics is that most studies have mainly focused on the young without sufficient attention paid to the elderly [5,18,19,20]. For an experimental design, only one single marker placed on the nipple is usually used to determine deformation of the entire breast [19,20,21]. The data accuracy is not convincing due to the complexity and various densities of the breasts [22,23]. The absence of information in the literature makes it challenging to well-establish a study of the three-dimensional (3D) strain of the breast skin of senior women during various movements.

In the last few decades, advanced techniques and computer-aided systems have been used to characterize the strain of breast skin. A common approach is to use laser body scanning to obtain 3D body anthropometric variables [24], which can meet the needs for ergonomic functional garment designs and fit optimization quickly and accurately. However, existing 3D models are unfortunately poorly validated and mainly limited to static body postures [25]. With the widespread application of 3D body scanning, scanning devices have been developed to conduct 4D scanning. They have integrated a fourth dimension, or time, which allows researchers to study dynamic breast deformation [26]. Furthermore, landmarks cannot be selected during scanning; thus, it is difficult to track a specific area or point. To observe the continuous changes in the strain of breast skin, the finite element (FE) method, which is a numerical simulation method, has been extensively used [27,28,29,30,31]. Although this method offers clear visualization and high accuracy, there are obvious shortcomings. For example, computation time is costly; non-convergence is another serious problem due to the complex geometries of the objects scanned; non-linear constitutive laws need to be considered for the elasticity of body parts, etc. Moreover, most of the FE models are individual-specific [28,29], which makes it difficult to draw a universal conclusion. Therefore, accuracy, continuous analysis, large datasets, and computation capacity all need to be balanced.

Recently, Shailaja et al. [32] proposed a computer-aided approach that combines machine learning (ML) with different technologies as an alternative solution, with the aim to develop an automatic and real-time function for multi-dimensional inputs and outputs. For instance, less than 1 s is required to simulate the mechanical behaviors of the breast under compression for clinical practice with a mean error under 3 mm by using extremely randomized trees [33]. One of the most popular ML algorithms is backpropagation artificial neural networks (BP-ANNs), which have proven to offer good adaptability and ease of generalization and can be widely used in various applications [34]. By learning the inputs and outputs as examples, BP-ANNs can change the variables to calculate the error iteratively until a configuration with a minimal error rate is obtained. The performance of BP-ANN models mainly depends on the architecture, a large enough database, and the adjustment of the connection weights [35]. On the contrary, highly multi-dimensional problems with a limited database are difficult to solve with the use of ML [36]. Consequently, there is the need to design a simplified system to extract the most significant indicators, and then predict new results that were previously unknown within a reasonable degree of accuracy.

In this regard, the purpose of this study is to develop a model order reduction approach to predict the real-time strain of breast skin during movement. This paper is structured as follows: information on the physical body part variables, and breast deformation and body motion variables are collected and presented in Section 2.1. The data are then used to train and reduce the input dimensions by sorting the significance of the variables, as shown in Section 2.2. The flow of the ANN predictions is provided in Section 2.3. The results and a discussion are provided in Section 3, including the results calculated from the gray relational analysis, performance of the ANN, and assessed ANN optimized models. This study provides a better understanding of the biomechanical properties of breasts of senior women, and a robust and interesting model that can inform sports bra designs for considerable improvement in wear comfort and physical health.

## 2. Materials and Methods

### 2.1. Participants

The study participants are 22 randomly recruited women who are on average 62 years old (they range from 54 to 69 years old) and found through online advertisements. Their average and standard deviation of height and weight are 154.8 ± 5.1 cm and 60.2 ± 5.6 kg, respectively. The selection criteria are: (1) bra cup size of 75C to 90F based on the metric sizing system [37], and (2) no previous breast surgeries. The experiment was approved by the Human Participants Ethics Sub-Committee of the first author’s university (Reference number: HSEARS20200205001). Written informed consent was obtained from all of the participants and the experimental details were provided and explained to them.

#### 2.1.1. Personal Body and Anthropometric Data Collection

To reduce the number of variables and simplify the ML model, the left side of the body is analyzed in this study, since the left and right sides are approximately symmetric. Thirteen physical variables of each subject were measured: age, height, weight, body mass index (BMI), bra underband size, bra cup size, underbust girth, breast circumference, left across-cup, which is the horizontal curve line passing through the nipple [28], acromion of the left shoulder to the nipple of the left breast (LBN), mid suprasternal notch (NPSN) to the LBN, LBN to the nipple of the right breast (RBN), and left shoulder angle [38]. These variables are shown in Figure 1. The corresponding value for each physical variable is shown in Table 1.

#### 2.1.2. Breast Motion Data Collection

The breast motion data of 22 mature female participants were collected by using a motion capture system (Eagle, Motion Analysis Corporation, Santa Rosa, CA, USA) at a sampling rate of 90 Hz. This system is popular and reliable for studies related to bra design and breast motion evaluation [39,40,41,42]. Thirteen spherical retro-reflective markers were attached on the left breast of the subject (see Figure 2). This study considered three degrees of freedom (x-direction, y-direction, and z-direction). Most of them have ptotic breasts; that is, their breasts have progressively descended and rest on their abdominal wall [43]. In other words, their aging and ptotic breasts could sag enough so that the nipples are below the inframammary fold if supportive garments are not worn. Thus, in this study, only the areas above the nipples are examined.

The participants were instructed to abduct their arm (Figure 2a), starting from the natural position where they are lowered at the sides to over their head to reach the highest position possible and hold for 3 s. Each step was repeated three times. A total of 15 retro-reflective passive markers (ɸ9 mm) were placed on the surface of the skin, and two on the lateral epicondyle and acromion, which were labeled O and A, respectively. The calculation of the arm abduction angle used in this study is shown in Figure 2b, in which Point B refers to a point in the horizontal line of Point O on the arm abduction plane. The angle between vector OA and vector OB quantities can be calculated by Function (1):(1)$$\theta =\arccos\left[\frac{\left(z_{2}-{z_{1}}^{2}\right)}{\sqrt{{\left(x^{2}-x^{1}\right)}^{2}+{\left(y^{2}-y^{1}\right)}^{2}+{\left(z_{2}-z_{1}\right)}^{2}}\times \sqrt{{\left(z_{2}-z_{1}\right)}^{2}}}\right]$$

The results from the experiments were used as preliminary inputs and outputs for the following machine learning model. The descriptions are provided in Table 2. LBN.MID is aligned with the midpoint between LBN and RBN and spaced at distance of 9 cm from the LBN. The two points spaced 3 cm apart and aligned between LBN and MID are labeled as IN1 and IN2, respectively. UP2 represents a distance of 9 cm from LBN in the vertical direction. The two points spaced 3 cm apart along this line are labeled MID3 and MID4, respectively. The middle line between LBN-MID and LBN-UP2 is defined as the UP2-UP1 line, where UP1 is 9 cm from LBN, and MID1 and MID2 are 3 cm from UP1. OUT3 is 5 cm from UP2 and 9 cm from LBN, while OUT1 and OUT2 are spaced 3 cm from each other on LBN-OUT3.

### 2.2. Gray Relational Analysis

Prior to building the BP-ANN model, the 14 variables used as inputs would lead to an overfitting problem and reduce the prediction accuracy. Therefore, the first task was to sort out several of the most significantly related variables. The gray relational analysis has been widely used in complex systems [44], decision-making problems [45,46], etc. The outstanding advantages of this method are that it does not require much data, involves typical distribution rules, and can explore the whole system by partially known information [44,47,48]. Therefore, the gray relational analysis was used here to calculate the significance of the order of the variables based on the gray relational degree. The gray relational degree is a mathematical model used to compare the geometric similarity of several different curves. A higher similarity means a larger degree of correlation and more significant influence. In this research work, the degree of correlation is obtained between the 14 inputs of the 13 body part variables plus one arm abduction angle and the output of 12 lines on the breasts. As previously stated, in total, 4400 frames were selected. Matrix A of the variables and angle with 4400 rows and 14 columns was built, while Matrix B of the displacement of the breast lines with 4400 rows and 12 columns was built. Then, the gray relational degree was determined. The detailed procedures are described as follows.

Determining reference sequence

All of the elements in the matrix are in the same order of magnitude, thus eliminating the dimensional impacts among the variables, which allows accurate comparisons of the calculations.

2.Calculating the gray relational coefficients

Each column in Matrix A denotes one sequence. Let a set of 14 sequences $\{x_{j}\left(k\right)\}=\left[x_{1}\left(k\right),x_{2}\left(k\right),x_{3}\left(k\right),\ldots x_{14}\left(k\right)\right],\;\;k=1,2,\ldots 4400$ be the reference sequence; let a set of 12 sequences in Matrix B $\{x_{i}\left(k\right)\}=\left[x_{1}\left(k\right),x_{2}\left(k\right),x_{3}\left(k\right),\ldots x_{12}\left(k\right),\right],\;\;k=1,2,\ldots 4400$ be the comparative sequence, so the gray relational coefficient is calculated by Function (2):(2)$$\xi_{\mathrm{ij}}(k)=\frac{\underset{i}{\min}\;\underset{k}{\min}\left|x_{j}(k)-x_{i}(k)\right|+\rho \underset{i}{\max}\;\underset{k}{\max}\left|x_{j}(k)-x_{i}(k)\right|}{\left|x_{j}(k)-x_{i}(k)\right|+\rho \underset{i}{\max}\;\underset{k}{\max}\left|x_{j}(k)-x_{i}(k)\right|}$$
where $\xi_{\mathrm{ij}}\left(k\right)$ is the gray relational coefficient of the association between the *i*-th comparison sequence and the *k*-th sample of the *j*-th reference sequence. The *p*-value ranges from [0, 1]; here, *p* is 0.5.

3.Calculating the correlation grade

The correlation grade determines the degree of correlation among each element in two systems, which can be calculated by Function (3):(3)$$r_{\mathrm{ij}}=\frac{\sum_{k=1}^{n}\xi_{\mathrm{ij}}(k)}{n}$$
where r_ij_ is the relationship between the i-th comparative sequence and the entire j-th reference column, n = 4400, k = 1, 2, …, 4400.

### 2.3. Back Propagation Artificial Neural Network Modelling

BP is a powerful optimization system to solve nonlinear problems with complex causal relationships. Hegazy et al. reported that a 3-layer neural network architecture can approximate any function mapping relationship [49]. Based on the results of the gray relational analysis, 8 factors with more influence on breast displacement were selected. Along with a BP ANN, the nonlinear relationship between the 8 selected predictive factors and 12 breast displacements was determined. To enhance the modeling performance, the size of the hidden layers and number of neurons need to be determined. Therefore, the feedforward net function was adopted to build the BP ANN model in which c_1_ was used as the network input, and b_1_ was used as the network output. The number of input neurons is 8, and output neurons is 12. The number of neurons in the hidden layer has a great impact on network performance. Too many neurons would cause over-fitting, while not enough would render the network fault-tolerant with low recognition ability. Refer to the previous empirical formula proposed by Zhang et al. [50]; the numbers of neurons in hidden layers were determined according to the following Function (4):(4)$$n=\sqrt{n_{i}+n_{o}}+a$$
where n, n_i_, and n_o_ are the number of neurons in the hidden, input and output layers, respectively. The a is a constant in the range of (1, 10). Because the number of input neurons is 8, and output neurons is 12, n_i_ is 8 and n_o_ is 12. Therefore, n can be calculated within the range of 6 to 15 with an equation.

As aforementioned, the selection of the activation function is important. Considering that the output of this paper is breast displacement, which could be any value; thus, the logsig and purelin functions were selected for the input-hidden and hidden-output layers, respectively [51], as follows, Functions (5)–(7):(5)$$f\left(y\right)=\mathrm{logsig}\left(y\right)=\frac{1}{1+e^{-y}}$$
(6)f(y)=y
where y is the weighted sum and presented as:(7)$$y=\overset{k}{\underset{j=1}{\sum }}w_{\mathrm{ij}}x_{j}+b_{j}$$
where k represents the number of processing elements in the previous layer, and $w_{\mathrm{ij}}$ and $b_{j}$ represent the weights and bias, respectively.

The BP ANN process was: (1) the initial weights and biases of the BP ANN were randomly set; (2) the weights among the neurons were adjusted according to the difference between the predictions and objective outputs; (3) the error signal was then sent back through the network and the weights were tuned accordingly, and the backward pass repeated until a set of optimal weights was found. The gradient descent backpropagation (GDBP) learning algorithm is the most common one in BP ANNs. However, the GDBP learning algorithm has the drawbacks of slow convergence, easily falls into the partial minimum, and training oscillation [52]. There are also several improved learning algorithms, including the momentum backpropagation (MOBP), variable learning rate (VLN) backpropagation, resilient backpropagation (RPROP), and conjugate and gradient backpropagation (CGBP). Therefore, these five learning algorithms have been used and then selected.

Based on previous results of the number of neurons in the hidden layers, which ranges from 6 to 15, five types of the aforementioned learning algorithms were used to investigate their fit performance. The number of neurons in the hidden layer was set to be between 6 and 15. There were a total of 50 different networks N_i_, where N referred to the network and i ranged from 1 to 50.

A three-layered BP ANN was built to predict the non-linear relationship between the 8 variables and 12 breast displacements by using MATLAB R2018a (The Mathworks, Inc., Natick, MA, USA). Initially, c_1_ and b_1_ were the training groups to train the 50 networks N_i_. After all the training was completed, c_2_ and b_2_ were used as test groups to test Ni.: c_2_ was inputted into 50 trained networks N_i_, and the network would calculate its trained weights to obtain the output value D_2_ (namely, the breast displacement predicted by the BP neural network). Finally, the error percentage between the target output D_2_ of each network N_i_ and the target output b_2_ was calculated to compare the prediction accuracy of each network N_i_. To obtain sufficient confirmation of the model capability, the following three evaluation techniques were used. The mean square error (MSE; Equation (8)) is usually used to provide a statistical measurement of the differences between the predicted results and the experimental results. The R correlation coefficient (Equation (9)) characterizes the correlation between the actual data and predicted data. The correlation coefficient obtained by the ANN model in MATLAB always refers to the Pearson correlation coefficient. The mean absolute percentage error (MAPE; Equation (10)) can intuitively interpret the relative error.
(8)$$MSE=\frac{1}{N}\overset{N}{\underset{t=1}{\sum }}{\left(actual_{t}-predicted_{t}\right)}^{2}$$
where $actual_{t}$ and $predicted_{t}$ represent the actual data and the predicted data; and t = 1, …, *N*, with *N* representing the sample number of the actual data.
(9)$$R=\frac{1}{N-1}\overset{N}{\underset{t=1}{\sum }}\left(\frac{actual_{i}-\mu (actual)}{\sigma (actual)}\right)\left(\frac{predicted_{i}-\mu (predicted)}{\sigma (predicted)}\right)$$
where σ(actual) and μ(actual) are the mean and standard deviation of the actual value and σ(predicted) and μ(predicted) are the mean and standard deviation of the predicted value, and *t* = 1, …, *N*, with *N* representing the sample number of the actual data.
(10)$$MAPE=\frac{1}{N}\overset{N}{\underset{t=1}{\sum }}\left|\frac{actual_{t}-predicted_{t}}{actual_{t}}\right|\times 100\%$$
where $actual_{t}$ and $predicted_{t}$ represent the actual data and the predicted data; and *t* = 1, …, *N*, with *N* representing the sample number of the actual data.

### 2.4. Particle Swarm Optimization

Kennedy and Eberhart introduced the concept of particle swarm optimization (PSO), which has been widely used for optimization because of the versatility of numerical experimentation [53]. It was inspired by natural animal social behavior, such as flocks of birds that are seeking food [54,55]. The population, called a swarm, refines its position to find the most promising region in a given search space. The individual in a population, which is called a particle, represents the candidate solution to solve the optimization problem. Each particle corresponds to a fitness value determined by the objective function. The particles change their velocities and positions to find the position with greater fitness. Each particle is influenced by the population, and an inertia weight is introduced to adjust the position of the particles. The best position memorized by each particle is called the local best-known position. The best position encountered by all particles that are memorized by the population is called the global best-known position.

The idea of using PSO to optimize a BP ANN is to obtain the best set of weights where the particles are moving to the optimal point, or the best solution. The advantages of the PSO are that it does not have conditional constraints, and at the same time, relies on a population of individuals to search for the adaptable information. Thus, by using a PSO-BP ANN model, not only would very strong global searching ability be realized, but also the BP ANN could be equipped with a strong local searching ability.

## 3. Results

### 3.1. Gray Correlation Grade Calculation for Variables Selection

In order to select the optimal variables for the following BP-ANN prediction, the gray relational analysis was performed to calculate the correlation grade among the independent variables of body variables, arm abduction angle, and dependent variables of breast displacement. Table 3 shows the result of the correlation grade of mapping one-to-one with independent and dependent variables. The first column presents the value of body variables including age, weight, height, body mass index (BMI), bra underband size, bra cup size, underbust girth, breast circumference, left across-cup, shoulder to LBN, left FNP to LBN, LBN to RBN, left shoulder angle, and arm abduction angle. The first row presents the value of LBN-UP1, UP1-UP2, UP2-UP3, LBN-OUT1, OUT1-OUT2, OUT2-OUT3, LBN-IN1, IN1-IN2, IN2-MID, LBN-MID4, MID4-MID3, and MID3-UP2. The results in Table 3 show that the arm abduction angle has the greatest influence on each breast displacement. The largest correlation grade was MID4-MID3 of 0.9, followed by LBN-MID4, IN1-IN2, LBN-IN1 of 0.86, and then IN2-MID of 0.85. While underbust girth, breast circumference, across-cup, shoulder to BN, FNP to BN, left BN to right BN, shoulder angle, and arm abduction angle have an average correlation between 0.5–0.6 for breast distance.

In conclusion, the arm abduction angle had a strong influence on the overall breast movement, followed by underbust girth, breast circumference, across-cup, shoulder to BN, FNP to BN, and left BN to right BN; and shoulder angle had less effect on breast skin displacement, and the remaining factors had little effect on breast movement.

### 3.2. Performance Assessment of Backpropagation Artificial Neural Network Modelling

#### 3.2.1. Overall Results of Backpropagation Artificial Neural Network Modelling

Once the most related variables were selected, the BP ANN models were built using the selected eight variables reported in 3.1. The total 4400 data was divided into 4300 training sets and 100 test sets. The test sets were used to validate the predicted results of the models. Finally, the BP ANN models, which consisted of one hidden layer with various neurons ranging from 6 to 15, the logsig algorithm as the training function, and purelin logarithm as a learning function were built for predicting the breast displacements.

The results of MAPE using several different BP methods are summarized in Table 4. It was clear that the minimum MAPE value was 13.1% of network N^30^, which proved that N^30^ had the optimal weights and bias among 50 networks. It displayed the performance that the predicted value reached 86.9% of the real value. The network structure of N^30^ had 12 neurons in its hidden layer, and its corresponding learning algorithm was the variable learning rate (VLN) backpropagation.

For clear observation, the MAPE results were drawn into a 3D graph as shown in Figure 3. The x-axis represented the 12 different breast displacement parameters, y-axis the 100 test samples, and z-axis the values of 12 breast displacement parameters. Most data points predicted by BP ANN were close to the experimental results, but several obvious errors can be found at distinct points on the standard horizontal base plane. The correlation R value for the training set is 0.89985, validation set 0.90019, test set 0.89886, and R-value for the overall training regression process was 0.89977.

#### 3.2.2. Optimization of Backpropagation Artificial Neural Network Modelling

With regards to the performance of the PSO-BP ANN, the R-value of the training set is 0.98907, validation set 0.98617, test set 0.9882, and R-value of the overall training regression process 0.98851. The MAPE results in 3D illustration are shown in Figure 4.

The training performance of MSEs of the BP ANN and PSO-BP ANN model are plotted in Figure 5. The error curve reached the lowest level during the training, and lower than that of the BP ANN and GA-BP ANN. The MSE of the measured and estimated values of the breast displacements is 11.52. It showed better validation performance after 199 iterations. The above analysis indicates that the PSO-BP ANN can predict breast movement well according to the 8 human body part variables in this study.

The derived weights and biases from the input layer to the hidden layer are listed in Table 5. The weights and biases from the hidden layer to the output layer are listed in Table 6. Based on the transmission process, the fixed weights and bias, calculation relationship between the 8 human body part variables, and 12 breast displacements can be calculated by Function (11):(11)b=w1×[logsig(w2×c+b1)]+b2
where b denotes the 12 breast displacements, c denotes the 8 body part variables, and logsig is the transmission function.

## 4. Discussion

The aim of this study is to determine key variables that are closely related to aging breast deformations during arm abduction and use these variables to accurately predict breast deformations. The clinical- or health-related impact of measuring the deformations of the aging breast during arm abduction helps to determine the magnitude of the tissue’s response to loading with low intensity but large motion range. As per Norris’ study in 2020, it was reported that strains on breast skin and other supporting structures were associated with the etiology of breast pain and skin damage [56]. Previous research works were conducted to study the strains of breast skin during walking or running and the correlation under various breast conditions [57,58]. However, some crucial factors such as body and anthropometric measurements were excluded. Thus, this study takes more factors into overall consideration, so that the results can be used to more precisely predict breast deformation.

### 4.1. Correlation Grade of Body Variables and Arm Abduction Angle on Breast Displacements among Older Women

In the first place, the gray correlation grade among body variables, arm abduction angle, and breast displacements were calculated. It is generally believed that when the gray correlation grade value is greater than 0.8, there is a significant influence relationship between the two factors. When it is less than 0.5, the influence relationship is not obvious [59]. Based on this result, the variables were arranged in order of largest to lowest. Underbust girth, breast circumference, across-cup, shoulder to BN, FNP to BN, left BN to right BN, shoulder angle, and arm abduction angle were the variables that have a correlation grade above 0.5. Underbust girth, breast circumference, and across-cup were usually used to determine the cup size [60]. Cup size and underbust girth were the most used variables to group participants for studying breast biomechanics, which may indirectly confirm that breasts with larger volume displaced increased static and dynamic breast skin strains [56,57]. However, from our study, shoulder to BN, FNP to BN, left BN to right BN, and shoulder angle were almost as important as underbust girth, breast circumference, and across-cup. Thus, these anthropometric variables also should be considered in further breast biomechanics studies.

While observing breast displacements, a large degree of correlation grade appeared in abduction and displacements. The largest correlation grade was the MID4-MID3 of 0.9, followed by LBN-MID4, IN1-IN2, LBN-IN1 of 0.86, then IN2-MID of 0.85. Considering the tendency of arm abduction was monotonically increasing, a larger value of displacements at those areas had the same varying tendency. Those areas can be divided into two parts, one part located at the middle part between two breasts and the other one near the nipple. It has been reported that most participants have the largest breast skin strain in upper breast areas longitudinally along the nipple’s vertical direction [60,61], and the results of this study in part agreed with previous research that the upper area displayed larger displacement. This phenomenon may result from the following reasons: firstly, the selected lines may be located at a direction perpendicular to Langer lines (lines reflect the direction of maximum skin tension) allowing skin extensibility. Secondly, studies have reported that the upper area had the thinnest thickness and greatest percentage change as age increased [61]. Additionally, the inner area also had a relatively larger correlation. Although studies seldom report on this phenomenon, this interesting finding may derive from the chest wrinkles and folds in the inner area developed from the aging process [62,63]. Those physiological changes in skin are directly related to the mechanical behaviors of the breast and should be paid more attention to as it is potentially relevant to skin damage and acceleration of breast deterioration.

### 4.2. Possibilities of Using Optimized Backpropagation Artificial Neural Network Modelling for Predicting Breast Displacements

From the results of BP ANN prediction, it was easily noted that the error curve was not smooth enough during the training process. This might be caused by major disadvantages of the BP ANN model, which are the slow convergence rate and easily trapped at the local minima. Thus, PSO was used to solve this problem. By randomly and iteratively changing the position of each particle in swarm, the weight can be updated. The repeat process of updating the weight of the network aimed to decrease the error of the current epoch. The particle can be placed to find the global optimal position with the lowest error. Once the acceptable error was achieved, the training process ended. In comparison with the former BP ANN models, the overall error of the optimized PSO-BP ANN was found to approximate the standard horizontal plane. Problems occurred in the BP ANN and GA-BP ANN in that some of the sample points far from the base plane had completely disappeared. Since both the R-value and MSE have been significantly improved, it potentially proved that the non-convergence of the individual networks or being trapped at the local minima during the BP ANN training process was solved by this optimization algorithm, and the network had jumped out of the local minima. In other words, the PSO-BP ANN had the advantages of both PSO and BP ANN, which were the strong global and local searching ability.

### 4.3. Study Limitations

This study selected the top eight correlated variables and proved the feasibility of using these variables to predict aging breast deformations. However, there are some limitations, for example; this study does not consider the biological tissues responses to different arm rotation angles. Previous studies usually measured the breast skin strains during running and walking, but those experiments were excluded in the present study due to the potential risks (breast pain, injury, muscle soreness). Moreover, the participants involved in this study were healthy senior women; we anticipate conducting an experiment on frail senior women or breast cancer survivors to avoid unwanted injury or skin damage.

## 5. Conclusions

Considering the limitations and difficulties during data collection and processing of information of the breasts of elderly women, it is necessary to build models to study breast-related biomechanical behaviors and accurately predict breast deformations. Thus, in this study, BP ANN and optimized PSO-BP ANN models were built to determine the relationship among the participant variables, arm abduction angle, and breast deformation.

Firstly, the dimensions of variables as model input were reduced by gray relational analysis from 12 to 8, as the underbust girth, breast circumference, left across-cup, shoulder to LBN, left FNP to LBN, LBN to RBN, left shoulder angle, and arm abduction angle showed a larger correlation coefficient on the breast displacements. Then, the BP ANN model and PSO-BP ANN were built to predict the 12 breast displacements based on the eight selected variables. The best solution for the network bias and weights was determined and within the acceptable accuracy, whose R-value achieved 0.99. The optimized PSO-BP ANN can be a supplementary approach for breast displacement predictions and assisting biomechanical analysis. Furthermore, this result is helpful for preventing older women from over-stretching breast skin and ensuring the ergonomic designs of functional clothes in a cost-efficient and accurate way. This study can potentially provide the basis for future research in sports science and sports bra design for senior women, which would contribute to the well-being of the female population globally.

## Figures and Tables

**Figure 1 ijerph-20-00468-f001:**
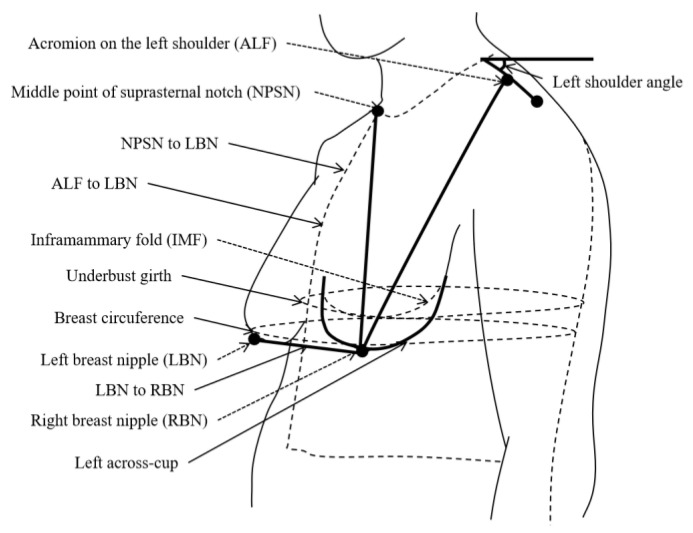
The anthropometric measurement of breast-related parameters.

**Figure 2 ijerph-20-00468-f002:**
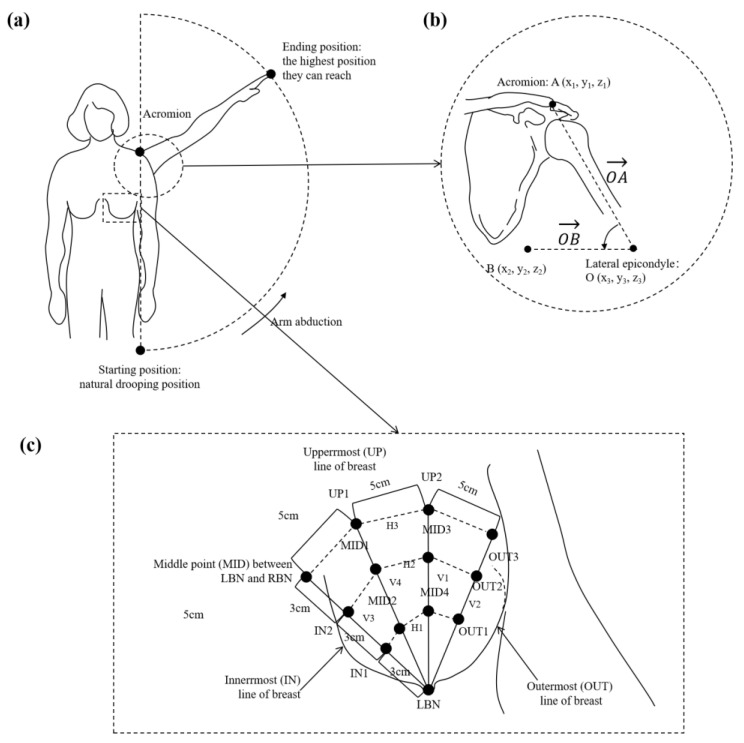
(**a**) The illustration of the posture during the motion data collection; (**b**) during the arm abduction process, the calculation of the arm abducting angle used in this study, which is the included angle between the vector OA and vector OB; (**c**) the definition of the markers on breast location.

**Figure 3 ijerph-20-00468-f003:**
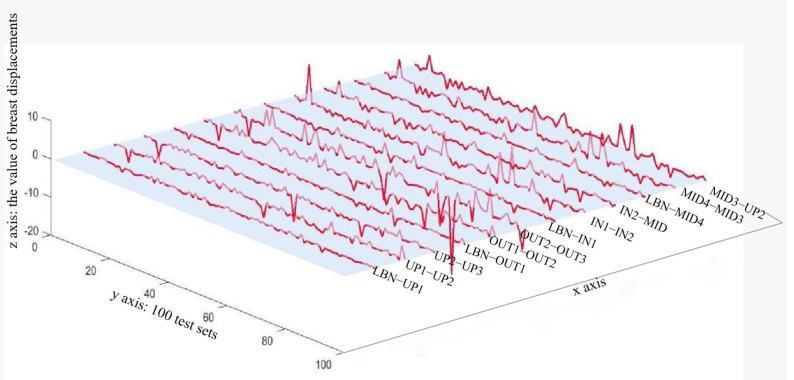
3D illustration of MAPE results predicted by BP ANN model.

**Figure 4 ijerph-20-00468-f004:**
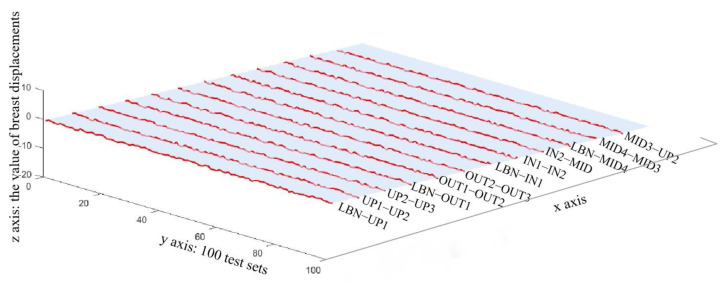
3D illustration of MAPE result predicted by PSO-BP ANN model.

**Figure 5 ijerph-20-00468-f005:**
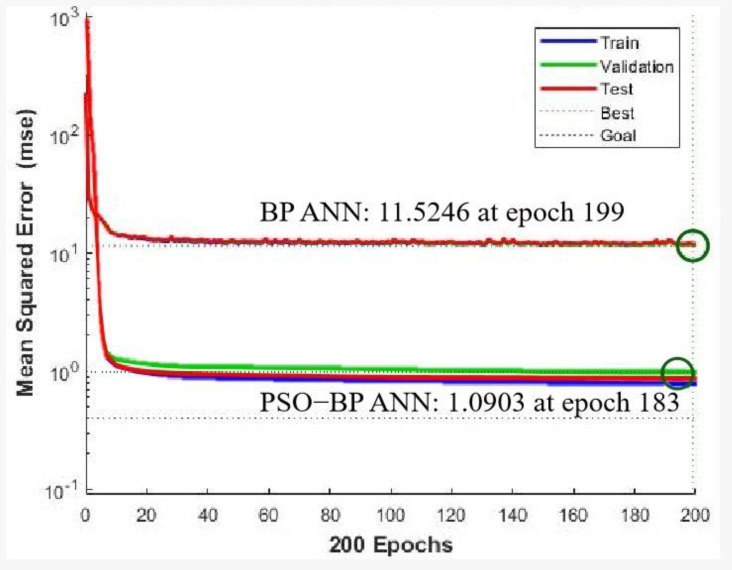
Training performance of the BP ANN and PSO-BP ANN models.

**Table 1 ijerph-20-00468-t001:** The value of body parameters for each participant.

Subject No.	Age	Height/cm	Weight/kg	BMI	Bra Cup Size	Underbust Girth/cm	Breast Circumference/cm	Left Across-Cup/cm	Acromion on the Left Shoulder to LBN/cm	NPSN to LBN/cm	LBN to RBN	Left Shoulder Angle/°
No. 01	54	161	61.2	23.6	C	80.8	94.3	19.8	25.5	20.5	19.4	26.2
No. 02	58	155	63.4	26.4	F	79.7	99.3	22.3	27.6	23.6	22.0	27.5
No. 03	58	153	58.0	24.8	E	83.2	102.0	22.0	25.3	22.1	19.3	25.2
No. 04	65	148	51.4	23.5	C	78.5	92.4	21.1	26.0	22.4	15.4	34.7
No. 05	67	155	67.4	28.1	D	83.5	99.0	19.6	24.5	21.6	19.0	28.1
No. 06	63	149	63.4	28.6	D	89.5	104.5	22.8	26.5	21.5	18.8	22.3
No. 07	63	158	60.2	24.1	C	75.6	88.0	19.6	24.0	19.0	14.9	30.3
No. 08	57	158	57.0	22.8	D	75.5	90.2	20.7	24.5	19.7	15.7	28.2
No. 09	59	163	71.1	26.8	D	83.0	97.0	23.4	28.5	24.3	19.0	26.4
No. 10	68	157	70.1	28.4	C	85.5	97.1	20.5	26.5	22.0	17.2	28.1
No. 11	66	154	54.2	17.2	D	75.0	91.0	20.2	25.0	21.5	15.3	26.0
No. 12	68	152	59.9	25.9	C	84.7	99.0	22.3	26.7	20.1	15.6	28.1
No. 13	58	160	55.5	21.7	C	75.0	86.6	18.6	23.3	18.8	18.0	30.4
No. 14	59	153	63.0	26.9	D	82.5	97.5	21.6	24.0	19.3	18.5	30.3
No. 15	59	144	57.6	27.8	C	80.9	94.8	21.6	25.5	21.0	17.6	31.2
No. 16	69	158	53.7	21.5	D	73.5	89.0	19.0	24.0	20.4	16.8	28.3
No. 17	57	147	57.0	26.4	C	81.7	94.7	20.2	25.0	19.0	14.6	30.9
No. 18	60	160	62.1	24.3	D	88.5	102.6	22.8	29.3	23.6	19.2	29.8
No. 19	66	148	53.1	24.2	C	78.7	90.0	20.7	24.7	20.6	16.3	29.3
No. 20	61	159	56.7	22.4	C	76.5	89.6	19.8	26.1	19.7	15.3	30.3
No. 21	60	160	69.5	27.1	D	87.2	101.2	23.0	28.2	21.5	17.9	27.7
No. 22	62	155	59.4	24.7	D	75.5	90.2	19.8	26.4	22.3	16.7	24.4

**Table 2 ijerph-20-00468-t002:** Preliminary inputs and outputs for machine learning model.

No.	Input	Output
1	Age	Distance between LBN and uppermost (UP) line of breast
2	Height/cm	Distance between UP1 and UP2
3	Weight/kg	Distance between UP2 and UP3
4	Body mass index	Distance between LBN and outmost (OUT) line of breast
5	Bra underband size	Distance between OUT1 and OUT2
6	Bra cup size	Distance between OUT2 and OUT3
7	Underbust girth/cm	Distance between LBN and innermost (IN) line of breast 1
8	Breast circumference/cm	Distance between IN1 and IN2
9	Left across-cup/cm	Distance between IN2 and middle point (MID) between LBN and RBN
10	Acromion on the left shoulder to left nipple (LBN)/cm	Distance between LBN and MID4
11	Middle point of suprasternal notch (NPSN) to LBN/cm	Distance between MID3 and MID4
12	LBN to right nipple (RBN)/cm	Distance between MID3 and UP2
13	Left shoulder angle/°	Angle between line from neck-shoulder point and horizontal plane
14	Arm abducted angle/°	Angle between the line from the armpit (hand in an inferior position) and midline of the body in the coronal plane

**Table 3 ijerph-20-00468-t003:** The correlation grade among 14 body parameters and the 12 breast displacements.

Correlation Degree Value	LBN- UP1	UP1- UP2	UP2- UP3	LBN- OUT1	OUT1- OUT2	OUT2- OUT3	LBN- IN1	IN1- IN2	IN2- MID	LBN- MID4	MID4- MID3	MID3- UP2
Age	0.27	0.34	0.32	0.33	0.33	0.27	0.19	0.33	0.30	0.23	0.22	0.35
Height	0.23	0.30	0.28	0.35	0.29	0.25	0.26	0.36	0.35	0.31	0.27	0.33
Weight	0.26	0.32	0.30	0.38	0.32	0.28	0.32	0.40	0.37	0.36	0.33	0.36
BMI	0.25	0.32	0.30	0.38	0.31	0.27	0.34	0.39	0.38	0.37	0.33	0.34
Bra underband Size	0.36	0.43	0.41	0.50	0.42	0.38	0.45	0.49	0.52	0.49	0.45	0.45
Bra cup size	0.20	0.21	0.21	0.26	0.24	0.23	0.32	0.26	0.28	0.30	0.35	0.21
Underbust girth	0.43	0.50	0.48	0.56	0.49	0.45	0.47	0.56	0.57	0.51	0.47	0.53
Breast circumference	0.40	0.46	0.44	0.55	0.45	0.42	0.53	0.56	0.58	0.57	0.52	0.48
Left across-cup	0.50	0.56	0.54	0.66	0.55	0.53	0.63	0.65	0.68	0.67	0.61	0.58
Shoulder to LBN	0.49	0.55	0.53	0.65	0.55	0.52	0.62	0.64	0.66	0.68	0.62	0.57
Left FNP to LBN	0.47	0.53	0.51	0.62	0.53	0.50	0.63	0.62	0.65	0.69	0.63	0.55
LBN to RBN	0.43	0.46	0.45	0.55	0.48	0.46	0.64	0.55	0.60	0.62	0.63	0.48
Left shoulder angle	0.45	0.51	0.49	0.53	0.51	0.45	0.46	0.54	0.55	0.47	0.47	0.52
Arm abducted angle	0.77	0.80	0.80	0.84	0.83	0.81	0.86	0.86	0.85	0.86	0.90	0.80

**Table 4 ijerph-20-00468-t004:** The MAPE (%) results of the BP ANN model using different learning algorithms.

Number of Neurons (Hidden Layer)	MAPE (%) by Various Learning Algorithms				
	GDBP	MOBP	CGBP	RPROP	VLN
6	N^1^(17.7%)	N^2^(15.2%)	N^3^(14.5%)	N^4^(14.5%)	N^5^(13.3%)
7	N^6^(19.6%)	N^7^(16.6%)	N^8^(14.0%)	N^9^(14.0%)	N^10^(13.6%)
8	N^11^(18.9%)	N^12^(17.7%)	N^13^(14.7%)	N^14^(14.7%)	N^15^(13.7%)
9	N^16^(17.5%)	N^17^(16.7%)	N^18^(14.5%)	N^19^(14.8%)	N^20^(13.7%)
10	N^21^(17.8%)	N^22^(15.3%)	N^23^(14.8%)	N^24^(14.8%)	N^25^(14.5%)
11	N^26^(18.5%)	N^27^(15.6%)	N^28^(14.6%)	N^29^(14.5%)	N^30^(13.1%)
12	N^31^(18.2%)	N^32^(16.2%)	N^33^(14.9%)	N^34^(14.9%)	N^35^(13.9%)
13	N^36^(19.7%)	N^37^(18.8%)	N^38^(15.9%)	N^39^(15.9%)	N^40^(13.8%)
14	N^41^(19.3%)	N^42^(18.2%)	N^43^(15.35%)	N^44^(15.2%)	N^45^(13.2%)
15	N^46^(18.9%)	N^47^(18.3%)	N^48^(15.9%)	N^49^(14.9%)	N^50^(13.8%)

**Table 5 ijerph-20-00468-t005:** The weights and bias from the input layer to the hidden layer.

w1								b1
−0.010	−1.048	1.582	−7.208	2.790	2.177	−0.536	1.289	−52.500
−0.197	−19.367	10.071	−6.369	13.647	19.055	−3.162	5.684	−75.619
−0.014	0.804	0.349	−3.800	−0.027	−0.098	3.652	−5.372	54.097
−0.342	2.342	−1.826	1.545	−3.742	2.360	−2.014	−0.347	63.730
−0.029	4.488	−3.997	1.586	−4.843	2.083	−0.566	0.776	51.040
−0.006	53.724	−41.630	−24.888	−5.534	−33.960	−33.537	51.446	3.809
0.004	9.841	−5.817	−13.680	−8.531	11.753	−2.689	2.603	−1.797
0.000	0.854	−0.451	3.798	−1.362	−2.483	0.950	0.900	−56.760
−0.001	3.257	−1.502	10.777	−3.417	−10.993	5.818	−3.216	−35.393
−0.002	−14.456	18.523	−8.315	−0.696	2.606	−11.888	−7.712	−13.755
−0.004	1.446	−9.972	−11.381	−1.969	33.103	−7.837	17.092	15.084

**Table 6 ijerph-20-00468-t006:** The weights and bias from the hidden layer to the output layer.

w1											b1
−4.355	−1.961	6.768	1.038	−13.321	−6.471	17.698	15.025	5.251	1.609	7.369	20.718
−8.956	2.835	23.007	2.499	0.307	−12.118	11.223	12.103	0.959	2.530	8.823	24.272
−10.458	6.771	31.326	2.567	6.966	−12.720	9.051	8.774	4.312	8.023	22.185	20.153
−5.199	2.925	−4.201	−0.526	−2.933	−5.716	10.163	4.422	14.135	4.688	17.431	22.513
−14.029	7.974	3.193	4.999	−3.178	−7.738	14.917	11.048	4.683	6.532	6.536	18.344
−14.104	16.504	−6.633	11.439	23.400	−7.142	6.071	−10.219	11.852	19.494	26.576	8.493
1.565	−2.581	−3.363	−0.683	−3.219	9.362	−2.693	−0.945	10.077	16.910	11.685	25.174
−0.634	−4.802	−6.711	−4.255	−3.403	−8.806	8.082	13.033	−0.356	2.650	5.280	30.560
18.614	−1.278	−5.582	−1.050	12.584	−2.331	−1.303	−9.887	25.511	11.855	27.711	8.702
3.481	−0.781	2.008	−1.307	−2.779	2.778	0.783	−3.568	10.910	9.893	12.617	26.067
2.499	−4.119	9.285	−5.471	6.988	−7.855	4.113	6.572	5.670	10.965	26.380	18.259
−3.797	−4.049	4.465	−2.045	0.101	−11.777	8.836	15.087	−0.798	3.695	10.170	27.366

## Data Availability

Not applicable.

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
