# Peer review of "A Systematic Analysis of 3D Deformation of Aging Breasts Based on Artificial Neural Networks"

_ijerph, 2022, doi:10.3390/ijerph20010468_

Round 1

Reviewer 1 Report

In their systemic analysis  of 3D deformation of aging breasts based on artificial neural networks Zhang et al. developed a model on 22 female patients to investigate the reduction approach to predict the real-time strain of breast skin of seniors during movement.

Although the study is written up well, I miss the real clinical or health-related impact. The authors mention that s a better understanding of the biomechanical prope ties of breasts of senior women, and a robust and interesting model that can inform sports bra designs for considerable improvement of wear comfort and physical health. I am not sure if physical healt can be increased through the presented results. 

Please explain were the benefit of the study is for patients.

Please critcally revise limitations section.

An Abbreviaition section should be included.

Is there any studies in relation to patients with breast cancer - if yes please include

Reviewer 2 Report

The authors perform a study on the deformations of aging breasts. This area is shown to be relevant; it is increasingly important to understand how breast skin behaves with age for the purposes of cosmetic or reconstructive surgery, and sports biomechanics.

Overall, the paper is interesting to read. Study of relevant literature is performed. The method is clearly described, the results are well presented, and necessary conclusions are made. Paper structure makes it easy to follow.

However, there are some drawbacks that should be considered to improve the quality of the paper before publishing:

1. Table 1 is showing 15 participants only. If the goal is to show data for all the women participating in the study, why is it only 15 people, and not 22? Table has duplicate data, 9 right-most columns are the same as first 9 columns. If those are removed, table will become easier to read. I would suggest having data for all 22 people and checking for duplicate information.

2. Why was a specific motion capture system chosen for the study (Eagle, Motion Analysis Corporation, US)? Please provide some explanation and mention other similar systems.

3. Why was the gray relational analysis used to determine the related variables, and not some other method? Please provide some advantages of this method over other methods with regards to your study.

4. Better to split longer paragraphs into several smaller ones based on the main idea, that would make the paper easier to read. Generally, paragraphs should be no longer than 5-8 sentences. For example, conclusion is a one large paragraph that can discourage readers from fully reading it.

5. Please include future possible research in conclusion, that will add value to the paper.

Once this is amended, the paper will be suitable for publication in this journal.

Round 2

Reviewer 1 Report

Improved. The study provides a better understanding about possibilites which can lead to breast pain. I suggest to accept the study for publication.